

# Prediction of temperature distribution in a furnace using the incremental deep extreme learning machine

Manli Lv[1,2], Jianping Zhao[1], Shengxian Cao[2], Tao Shen[3] and Zhenhao Tang[2]

[1] College of Computer Science and Technology, Changchun University of Science and Technology, Changchun, China
[2] School of Automation Engineering, Northeast Electric Power University, Jilin, China
[3] Harbin Boiler Company Limited, Harbin, China

## ABSTRACT

In this article, a data-driven model based on the incremental deep extreme learning machine (IDELM) algorithm is proposed to predict the temperature distribution in the furnace. To this end, computational fluid dynamics (CFD) simulations are carried out first to get temperature distributions under typical working conditions. Based on the air distribution mode, the simulation results are divided into six subclasses. Then the K-means clustering method is applied to find out the benchmark working condition of each subclass. Moreover, the random sampling method is used to extract samples to reduce computational complexity. Modeling inputs are selected according to the CFD boundary conditions and combustion mechanisms, and data sets are reconstructed based on the increments of each actual working condition from the benchmark working condition. Finally, an IDBN-based prediction model is built in each subclass. The experimental results show that the IDBN-based model has a promising predictive ability with less than 11% symmetric mean absolute percentage error.

## INTRODUCTION

Aiming at reaching the goals of the "Carbon peak and carbon neutrality" policy, clean sources of energy have been greatly developed in the past few decades. In this regard, numerous optimizations have been proposed in the field of power generation, which account for 44% of total carbon emissions (*Wang, Guo & Chen, 2021*). Despite the rapid development of clean sources of energy, coal-fired power generation still accounted for 71.13% of the total electricity generation in 2021 and it is still considered a stable source to generate electricity (*Li et al., 2021*). Pulverized coal combustion in thermal power plants is a multivariable coupled system that includes complex physical and chemical reactions. Studies show that the temperature distribution of the furnace affects the combustion stability and the generation of pollutants and unburned carbon losses. Accordingly, it is considered the main indicator of the combustion state. The monitoring and control of the temperature distribution can stabilize and optimize combustion, and prevent slagging and

Corresponding author
Jianping Zhao, zjp@cust.edu.cn

local overheating. It is worth noting that considering the high temperature of the furnace and the necessity to monitor the furnace status rapidly, most temperature and speed sensors cannot be directly installed in the furnace. Currently, the conventional methods for measuring the furnace temperature are sparse temperature point measurements. However, this method does not reflect the temperature distribution and does not support real-time 3D visualization of the furnace temperature. Therefore, it is of significant importance to investigate new schemes and resolve shortcomings in this regard.

Research of furnace temperature distribution can be mainly classified into two categories, including direct CFD simulation and indirect methods. With appropriate simplifications, CFD simulation can solve partial differential equations of the combustion process and two-phase flow numerically. The flow field, temperature field, and properties of combustion products can be obtained (*Dindarloo & Hower, 2015*). Based on CFD simulations results, the relationships between the temperature distribution and operating parameters are analyzed. Accordingly, the influence of different parameters on the combustion process can be analyzed, including the boiler load (*Xu, Azevedo & Carvalho, 2001*; *Laubscher & Rousseau, 2019*), burner arrangement and tilt angle (*Choi et al., 2020*; *Tan et al., 2017*), separated over-fire air (SOFA) ratio and SOFA location (*Ma et al., 2015*), yaw and tilt angles (*Jin et al., 2021*), distribution modes [11], secondary air boundary conditions (*Zadravec, Rajh & Kokalj, 2022*), and air staging combustion (*Zhang et al., 2015*; *Wang & Zhou, 2020*). With the development of new sources of energy and deep peak-load regulations, it is necessary to modify conventional thermal power units to work under ultra-low loads. In this regard, CFD simulations show that operating conditions at lower boiler loads considerably affect the flow and temperature fields and the concentrations of combustion products (*Chang, Wang & Zhou, 2022*; *Belosevic et al., 2019*; *Zhao et al., 2018*; *Yuan et al., 2019*). The temperature distribution of CFD can be used to the combustion stability monitor under ultra-low loads. Recently, the combustion simulation was coupled with steam generation model to obtain the flow behavior in the combustion chamber, the steam generation, and distribution (*Mahvelati et al., 2022*). Then an intelligent algorithm was used to improve the simulation accuracy of CFD (*Secco et al., 2015*; *Debiagia et al., 2020*). It was found that CFD simulations can be applied to simulate the macroscopic phenomena in the furnace and complex reactive flows. However, CFD is based on iterative methods to solve the partial equations and simulate the physical and chemical processes, which is a time-consuming process and cannot meet the requirements of real-time predictions. Furthermore, most analyses involved a few operating parameters, which do not correspond to multi-variable operating conditions in the actual field.

The second category is indirect combustion detection, which is the inversion or reconstruction of the 2D and 3D distribution of the temperature field using optical imaging or acoustic measurement methods. The optical method is based on the principle that objects with different temperatures have different radiation wavelengths. In this regard, heat-sensitive images of the flow field are captured by a CCD camera (*Zhou, Han & Sheng, 2002*; *Zhou et al., 2005*). Then the Monte-Carlo method is applied to calculate the radiation intensity and establish the relationship between the radiation flame image and the temperature distribution. Tikhonov regularization and its improved algorithm

and the least square QR decomposition (LSQR) method are the most widely used methods to reconstruct the temperature field. Recently, flame light field imaging was proposed as a new type of flame detection method (*Liu et al., 2012*). In this radiation-based method, the reconstruction technique was developed on the backward Monte-Carlo methods. The reconstruction matrix equations are solved using the LSQR method (*Li et al., 2019*). The acoustic temperature field reconstruction method is based on the principle that the propagation velocity of the sound wave is different at different temperatures. The inversion algorithms in this regard include line-integrated measurements (*Barth & Raabe, 2011*), ART iteration (*Ma, Liu & Cao, 2019*), radial basis function approximation polynomial (*Kong et al., 2020a*), kernel regression model (*Kong et al., 2020b*), and artificial neural network (*Jeong et al., 2021*). *Zhou, Dong & Zhao (2020)* combined the reflective sinusoidal radial basis function and QR decomposition method and proposed a temperature field reconstruction algorithm to solve the low accuracy problem of the temperature field edge reconstruction caused by the traditional acoustic temperature measurement algorithms. Accordingly, high-accuracy reconstruction of two-dimensional temperature fields was achieved. It is worth noting that both the optical image radiation method and the acoustic wave method can be applied to monitor the temperature field of the furnace under ideal conditions. However, both methods require additional equipment to be installed in the furnace. In such a high-temperature and complex furnace, there are uncertain factors such as abrasion and ash deposition that affect the measurement accuracy. Considering the interference originating from other devices and human measurement errors, large errors in the temperature field inversion are unavoidable in single measurement methods.

With the rapid development of computer technology and artificial intelligence technology, data-driven modeling based on deep learning has been widely used to predict the concentration of NOx (*Xie et al., 2020*; *Kang et al., 2017*), unburned carbon (*Dindarloo & Hower, 2015*), and predict thermal efficiency (*Yan, Wza & Xi, 2019*; *Ren, Zhang & Zhang, 2019*) in thermal power plants. At present, the common algorithms in parameter prediction of thermal power plants include artificial neural network (ANN), support vector machine (SVM), extreme learning machine (ELM), deep neural network (DNN) deep belief network (DBN) and long short-term memory network (LSTM). Although remarkable achievements have been obtained, these methods have not been used to model furnace distribution parameters. Based on the performed literature survey, it was intended to propose a novel data-driven model based on the incremental deep extreme learning machine (IDELM) algorithm to predict the furnace temperature distribution. To this end, furnace temperature distribution under typical working conditions was calculated using CFD simulation. Then different subclasses were defined in data-driven modeling according to the combustion air distribution mode. The K-Means clustering method was adopted to find out the benchmark working condition of each subclass and typical samples were extracted by random sampling. Finally, the IDBN-based prediction model was built in each subclass. The performance of the proposed model was further analyzed compared with other algorithms.

The main contributions of this study are as follows:

(1) CFD simulation results and boundary conditions are used as the data sets for furnace temperature distribution modeling, realizing the combination of mechanism modeling and data-driven modeling.

(2) For discrete variables such as the combustion air distribution mode and the coal mill operation mode, special treatment is performed to accommodate data-driven modeling. K-Means clustering is used to find the benchmark conditions and random sampling is applied for representative sample extraction to achieve data set reconstruction.

(3) Based on the incremental changes of input and output of typical and benchmark conditions, the IDELM algorithm is proposed to predict the furnace temperature distribution.

This article is organized as follows: The research object is introduced in 'Boiler Description and Overall Modeling Framework'. Then CFD simulations under typical operating conditions and the preparation of datasets are presented in 'CFD Simulation'. The modeling of the furnace temperature distribution based on the IDELM is displayed in 'Modeling the Temperature Distribution', and the results of the experimental analysis are presented in 'Results & Discussion'. Finally, the main conclusions are summarized in 'Conclusion'.

## BOILER DESCRIPTION AND OVERALL MODELING FRAMEWORK

### Boiler description

In the present study, a 350 MW supercritical coal-fired boiler is selected as the research subject. The boiler has a $\pi$-shaped arrangement with a single furnace chamber and a double flue. The furnace is 58.3 m high and has a cross-sectional area of 14.627 m $\times$ 14.627 m. The horizontal flue is 5.32 m long and the depths of the front and rear tail flue are 6.05 m and 6.82 m, respectively. A new type of tangential combustion is adopted in the boiler. Six layers of pulverized coal air chambers (A~F) are distributed in the main combustion area of the furnace, and each layer is arranged with four horizontal pulverized coal nozzles on the four walls of the water-cooled wall. Moreover, eight layers of the secondary air (AA, AB, BC, CC, DD, DE, EF, and FF) and four layers of separated over fire air (SOFA1~SOFA4) enter the furnace through the nozzles in the four corners of the chamber. The boiler structure and the burner arrangement are schematically shown in Fig. 1.

The main operating parameters at the rated power of the boiler are shown in Table 1.

## CFD SIMULATION

Figure 1 shows that the computational domain includes the furnace chamber, burner, SOFA nozzles, and horizontal and vertical flues. Firstly, the geometric model was established in the SolidWorks platform, and then the model was meshed using the ICEM preprocessor. In order to ensure calculation accuracy, structured hexahedral and refined unstructured meshes were used for the furnace body and the main combustion area, respectively. The grid system was simulated using 2.5, 2.8, and 3 million meshes, and the average temperature along the height of the furnace chamber was used as the indicator. Based on the performed

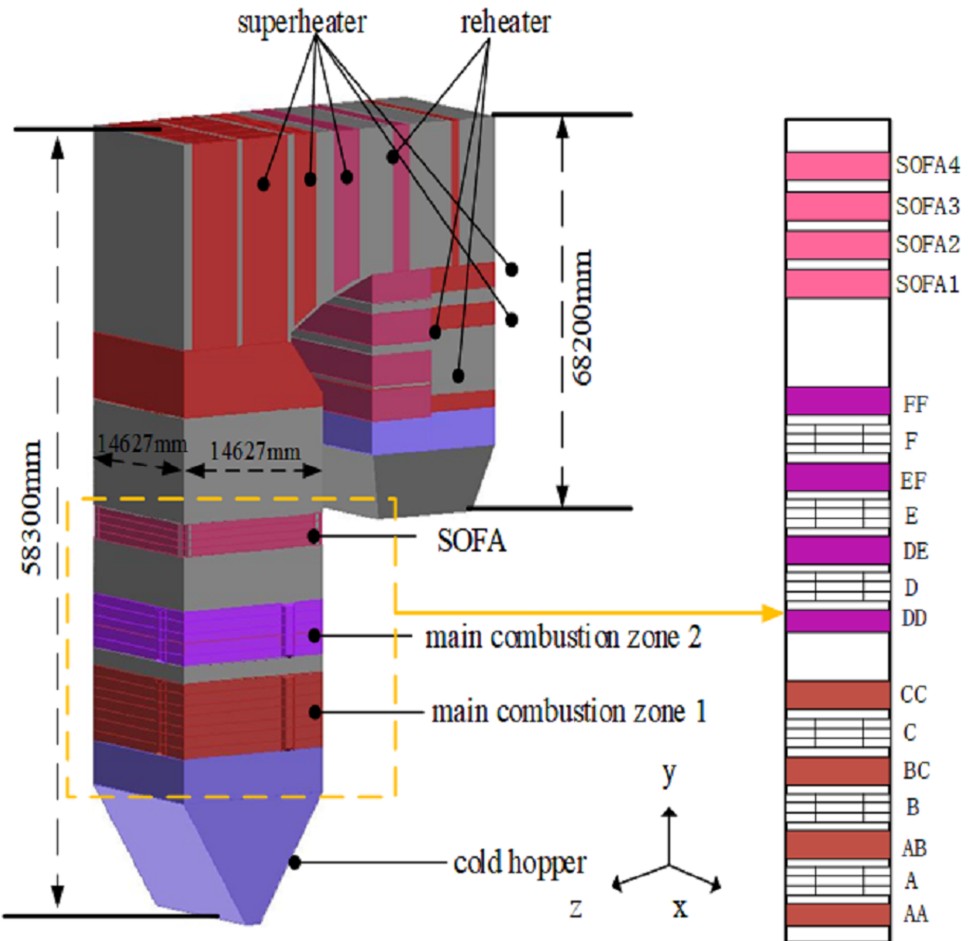

**Figure 1  The overall boiler structure and burner arrangement.**

**Table 1  Main boiler operating parameters at the rated power.**

| Parameter | Values | Unit |
|---|---|---|
| pulverized coal | 53.75 | kg/s |
| total air | 370.64 | kg/s |
| average excess air coefficient | 1.20 | – |
| primary air | 110.07 | kg/s |
| second air | 260.54 | kg/s |
| SOFA | 111.18 | kg/s |
| primary air temperature | 65.0 | °C |
| secondary air temperature | 356.0 | °C |

grid independence test and balancing simulation accuracy and computational speed of the numerical simulation, a grid with 2.8 million meshes was selected to simulate the combustion process.

**Table 2  Analysis of the pulverized coal.**

| Ultimate analysis (%) | | | | | Proximate analysis (%) | | | | LHV (ar) |
|---|---|---|---|---|---|---|---|---|---|
| C | H | O | N | S | Moisture | Ash | Volatile | Fixed carbon | Qnet |
| 44.82 | 2.68 | 10.26 | 0.52 | 0.13 | 31.75 | 9.84 | 24.78 | 33.63 | 16310 |

In the present study, Fluent 15.0 software was applied to simulate the combustion process. The gas-phase turbulence was calculated using a Realizable k-$\varepsilon$ model, which is closer to Reynolds averaged Navier–Stokes equations. The gas-solid two-phase flow is calculated using the stochastic tracking model in the Euler–Lagrange method. The combustion process includes the coal devolatilization, volatile combustion and char combustion. Coal devolatilization is modeled by a two-step competitive reaction model, volatile combustion is modeled by a non-premixed combustion model, and char combustion is described by a diffusion/kinetic model. The discrete ordinates (DO) model is used for the radiation. The coal quality is regarded as invariant during the simulation. The ultimate and proximate analysis results are presented in Table 2.

To evaluate the accuracy of the CFD simulation, the results at 100% load were compared with experimental data under the same conditions. In this regard, furnace exit gas temperature (FEGT), platen-superheater bottom gas temperature (PBGT), economizer exit temperature (EET), and oxygen content of outlet flue gas (O2%) were compared. Table 3 shows that the absolute error of FEGT is 39.55 K, which is equivalent to 3.11%. Furthermore, the absolute error of PBGT is 8.25K and the relative error is 0.52%. The relative error of EET and O2 concentration at the boiler outlet is 4.37%. The performed analyses demonstrate that the CFD model can be applied to accurately simulate the combustion.

CFD simulations are carried out for different loads, burner arrangement modes, secondary air distribution modes, SOFA distribution modes, and burner tilt angles. More specifically, 120 operating conditions were simulated the temperature distribution and the concentration of combustion products were obtained.

# MODELING THE TEMPERATURE DISTRIBUTION

## Framework of IDEM modeling

Modeling the temperature distribution in the furnace mainly consists of three steps, including CFD simulation, data sets classification and reconstruction, and IDELM modeling. The overall modeling process is presented in Fig. 2.

Step 1: Input parameters and boundary conditions are set according to the type of boiler and the unit load. Then CFD simulation is carried out to simulate the combustion s and flow process.

Step 2: Based on the secondary air distribution mode, the CFD datasets are divided into 6 subclasses. The K-Mean clustering method is used to find the benchmark working condition of each subclass. Representative samples of each working condition are selected and then the dataset is reconstructed using data obtained by solving the corresponding increments between other working conditions and the benchmark.

**Table 3  The comparison between CFD simulation and site values.**

|  | CFD simulation | site values | Relative error |
| --- | --- | --- | --- |
| FEGT (K) | 1311.7 | 1272.15 | 3.11% |
| PBST (K) | 1592.9 | 1601.15 | 0.52% |
| EET (K) | 719.5 | 695.15 | 3.5% |
| $O_2$ (%) | 3.94 | 4.12 | 4.37% |

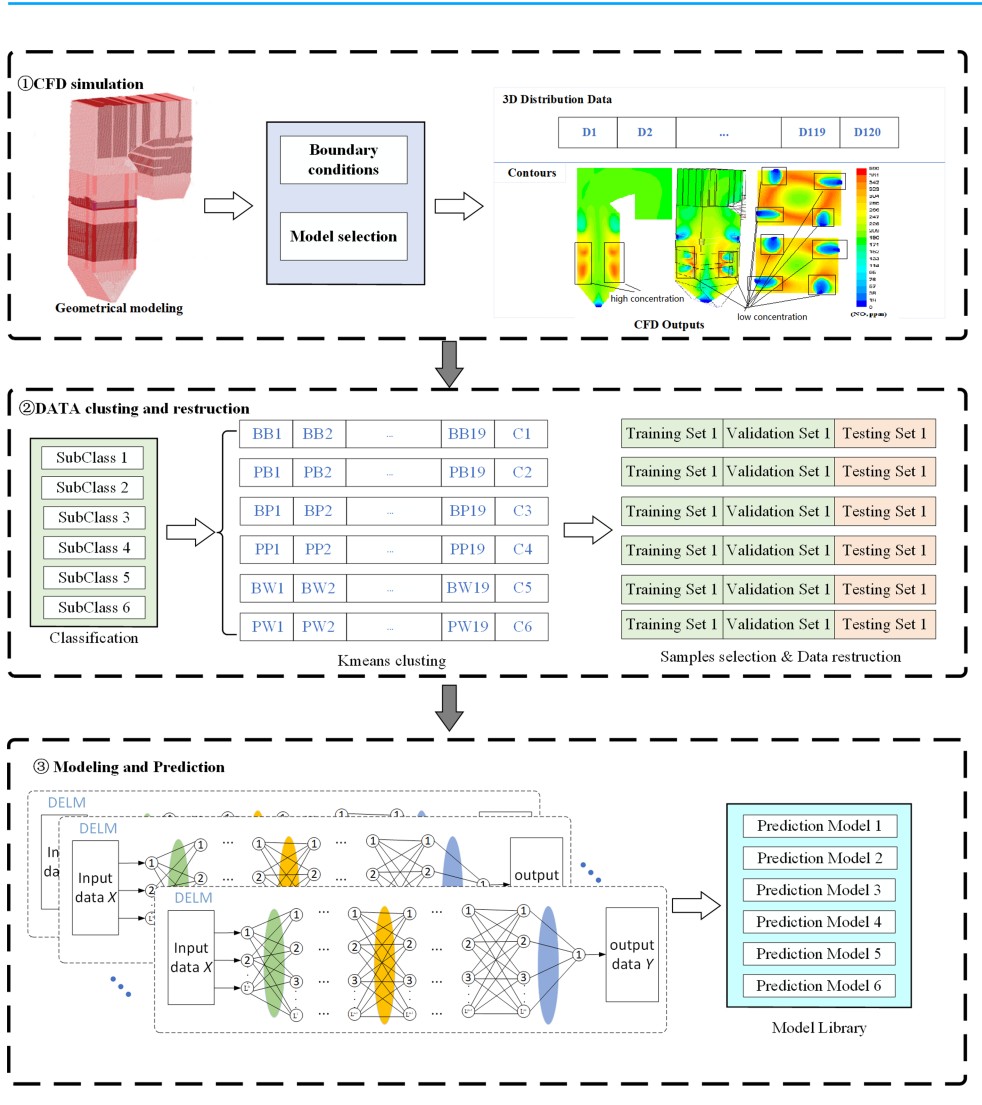

**Figure 2  Flow chart of IDELM prediction model.**

Step 3: Based on the reconstruction datasets of temperature increment of each working condition, prediction models of each subclass are established based on the DELM.

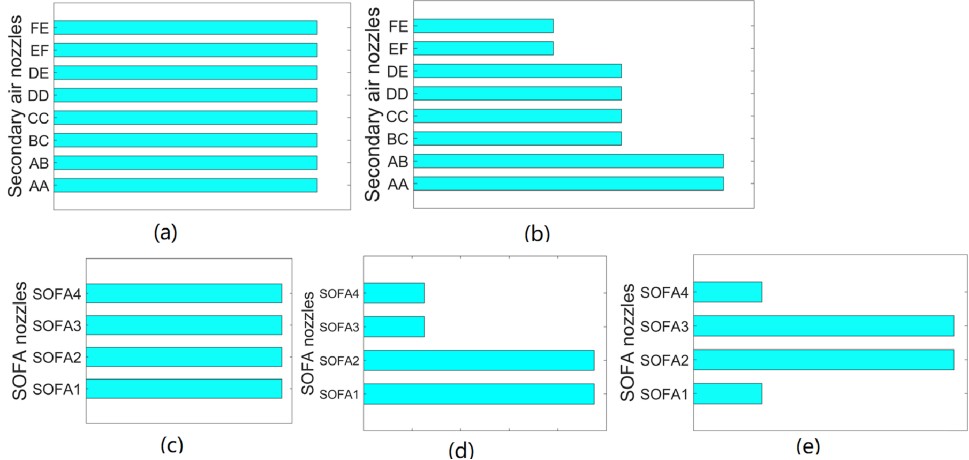

**Figure 3** **Air distribution mode of unit.** (A) Balanced mode secondary air dampers openings (B) Pagoda mode secondary air dampers openings (C) Balanced mode SOFA dampers openings (D) Pagoda mode SOFA dampers openings (E) Waist mode SOFA dampers openings.

**Table 4** **Classification of working conditions.**

| Classification | Secondary air distribution | SOFA air distribution | Datasets |
| --- | --- | --- | --- |
| Subclass 1 | Balanced | Balanced | BB1 ~BB20 |
| Subclass 2 | Pagoda | Balanced | PB1 ~PB20 |
| Subclass 3 | Balanced | pagoda | BP1 ~BP20 |
| Subclass 4 | Pagoda | pagoda | PP1 ~PP20 |
| Subclass 5 | Balanced | Waist drum | BW1 ~BW20 |
| Subclass 6 | Pagoda | Waist drum | PW1 ~PW20 |

## Classification of the working conditions

There are not only continuous variables among the inputs of the CFD model but also variables such as burner arrangement mode and distribution mode of the air dampers that have discrete nature. These variables affect the location of the center of the combustion flame, the temperature distribution, and the combustion products.

Although the volume of secondary air and SOFA are different under different loads, the flame center position and temperature distribution in the furnace are generally similar under the same air distribution patterns. Figure 3 indicates that there are two kinds of secondary air distribution: balanced mode and pagoda mode. Moreover, there are three SOFA distributions, including balanced mode, pagoda mode, and waist drum mode. In order to cover different air distribution modes, 120 working conditions are divided into six subclasses, in which each subclass has 20 datasets. The subclasses are listed in Table 4.

## Selecting clustering centers

The training sets for each subclass is limited. In order to improve the prediction accuracy, a benchmark working condition was set for each subclass. Then the increment-based

**Table 5  Clustering centers of each subclass.**

| subclass1 | subclass2 | subclass3 | subclass4 | subclass5 | subclass6 |
|-----------|-----------|-----------|-----------|-----------|-----------|
| BB10 | PB10 | BP10 | PP10 | BW10 | PW10 |

method was used to construct the temperature distribution model. Considering the high dimensionality, large number, and complex nature of the simulated experimental data, the K-means data clustering method was applied to select clustering centers. This method is an iterative clustering algorithm, which continues iterative calculations until the criterion function converges.

For each subclass, the temperature field data obtained from CFD simulation is extracted, and the distance function is used as the evaluation index for the similarity measure of the K-means clustering algorithm. Based on the minimum Euclidean distance between the benchmark working condition and other working conditions of this subclass, the benchmark working condition for each subclass is obtained. This can be mathematically expressed as follows:

$$D = \sum_{j=1}^{k} \sqrt{\sum_{i=1}^{n}(x_i - y_i)^2} \qquad (1)$$

Where, $k$ is the number of cluster centers, $x_i$ is the temperature distribution value in the benchmark working condition, $y_i$ is the temperature distribution value in other working conditions, and $D$ is the sum of the Euclidean distance.

The final cluster center selection is shown in Table 5. The clustering results show that the benchmark working conditions of the six subclasses are 75% load, BCDEF mill operation, and SOFA is 0 tilt angle.

## Data preprocessing

The studied boiler is equipped with six coal mills, and the pulverized coal enters the furnace through the burner nozzles on the four walls of the boiler. Three types of mill operation modes are designed for each load. Figure 4 shows the temperature distribution of YOZ planes in different coal mill operation modes at 100% load. Fig. 4A the case, in which ABCDE mills operate. In this case, the flame center is higher than others, while the temperature of the cold ash hopper is the highest. However, the lowest temperature occurs in the SOFA area. In Fig. 4B, the mill distribution range is large so the cold ash hopper area temperature and SOFA take the temperature between the middle. Fig. 4C shows that the flame center moves upward, the high-temperature area is large, and the highest temperature occurs in the SOFA area. Meanwhile, the lowest temperature occurs in the cold ash hopper area. It is inferred that the coal mill operation mode significantly affects the temperature field distribution.

The six mills in the actual plant are arranged at different furnace heights, but such discrete variables cannot be used as input for data-driven modeling. In order to resolve this problem, the coal mill operation mode is converted into a flame center position correction

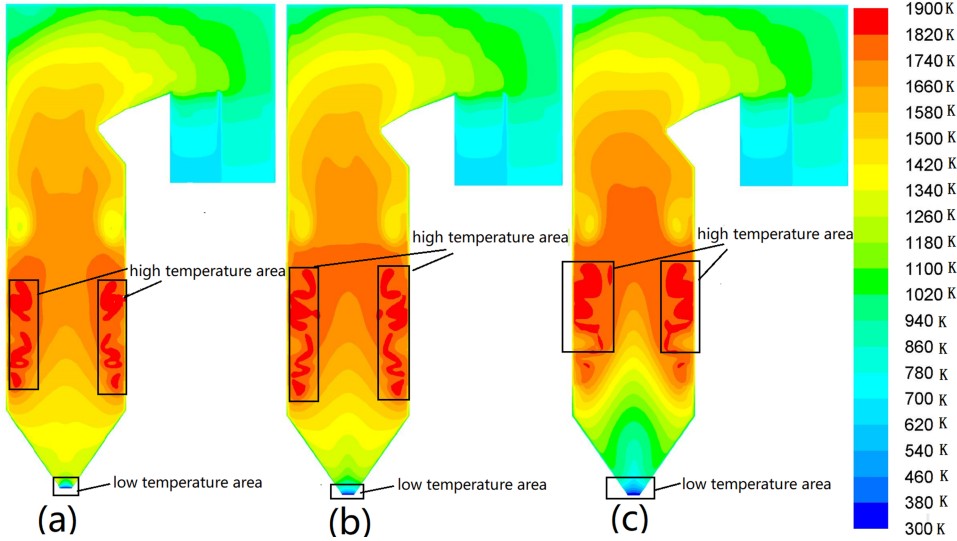

**Figure 4** Temperature distribution contours on YOZ planes in different coal mill operation modes. (A) ABCDE coal mill, (B) ABCDEF coal mill, (C) BCDEF coal mill.

factor.

$$M = A - B(x_r + \Delta x) \tag{2}$$

where M is the parameter that reflects the effect of the relative position of the highest temperature along the furnace height. *A* and *B* are empirical coefficients that depend on the fuel type and furnace structure, $\Delta x$ denotes the relative position correction value of the highest point of the flame (*Li, Yan & Liu, 2017*). $X_r$ is the relative height of the burner, which can be calculated in formula (3).

$$x_r = \frac{\sum n_i B_j H_{ri}}{H_L \sum n_i B_i} \tag{3}$$

where $H_L$ is the height of the furnace chamber, that is, the height from the bottom of the furnace or the middle plane of the cold ash hopper to the middle of the furnace exit smoke window middle height of the furnace chamber; $H_r$ is the height of the burner arrangement, that is the height of the burner axis from the middle plane of the cold ash hopper; $H_L$ is the amount of coal burned corresponding to the burner; $H_{ri}$ is the height of the burner arrangement corresponding to the layer; $n_i$ is the number of burners in the layer.

The variables in the 3D data set have different orders of magnitude. Table 6 lists the value range of each parameter.

The min-max normalization is carried out to preprocess the data. This can be mathematically expressed as follows:

$$x_i^* = \frac{x_i - x_{\min}}{x_{\max} - x_{\min}} \tag{4}$$

where $x_i$ is the original value, $x_i^*$ is the normalized value, $x_{max}$ and $x_{\min}$ $x_{\min}$ is the maximum and minimum value.

**Table 6  Simulation variables range.**

| Number | Variables | Unit | Value range |
|---|---|---|---|
| 1 | unit load | *MW* | [175, 350] |
| 2 | total air volume | *t/h* | [603, 1305] |
| 3 | total coal volume | *t/h* | [90, 212] |
| 4 | primary air volume | *t/h* | [218, 452] |
| 5 | secondary air volume | *t/h* | [330, 897] |
| 6 | SOFA dampers opening | % | [5, 100] |
| 7 | secondary air dampers opening | % | [10, 90] |
| 8 | secondary air temperature | ° C | [320, 366] |
| 9 | *x* coordinates | *m* | [−7.62, 7.62] |
| 10 | *y* coordinates | *m* | [6.5, 64.8] |
| 11 | *z* coordinates | *m* | [−25.51, 7.62] |
| 12 | temperature in furnaces | *K* | [300, 1900] |
| 13 | correction factor M | – | [1.03, 1.69] |

## Reconstruction of the dataset

The training data in machine learning should be selected in a way to balance the trade-off between computational complexity and accuracy. Based on the mesh independence test, a model with 2.8 million meshes is selected in the simulation, so each dataset has 2.8 million data. However, the huge dataset in data-driven modeling will increase the information redundancy and model complexity, thereby reducing the computational efficiency. Therefore, information extraction is the main challenge for a data-driven model.

The process of sample selection and dataset reconstruction is shown in Fig. 5. For 20 typical working conditions of each subclass, the increment between the data of 19 working conditions and the corresponding variables of the benchmark working condition is firstly calculated to obtain 19 new increment data sets. Similar to cross-validation, one dataset is selected as the test dataset, and the remaining 18 newly constructed datasets are randomly sampled. In the present study, 50,000 samples with the same spatial location are selected for each working condition. Finally, the selected samples are reconstructed into one dataset as the training and validation set.

## IDELM modeling

Deep extreme learning machine (DELM), also known as multi-layers extreme learning machine (ML-ELM), is a deep neural network formed by stacking multiple extreme learning machine autoencoder (ELM-AE). Its structure is shown in Fig. 6. In this method, ELM-AE is initially used as the basic unit for unsupervised learning to train and learn the input data. Then input weights and bias (W, b) of ELM-AE are randomly generated and the implied layer matrix H is formulated in the form below:

$$H = g(WX + b). \tag{5}$$

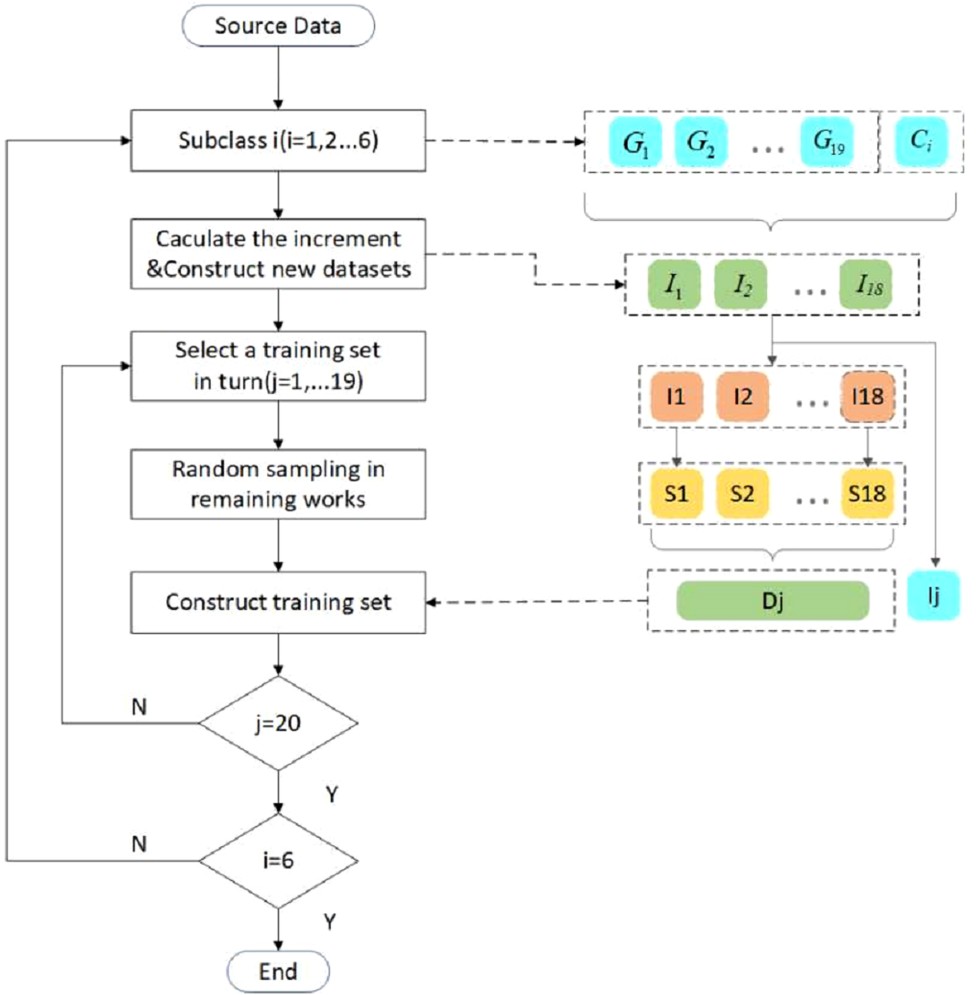

**Figure 5 Flow chart of sample selection and dataset reconstruction.**

The loss function of the ELM network is defined as follows:

$$\min L_{ELM-AE} = \frac{1}{2}\|\beta\|^2 + \frac{C}{2}\|X - H\beta\|^2. \tag{6}$$

DELM adds the restriction of the output weight regular term, which can prevent overfitting. The output weights $\beta$ of ELM-AE can be calculated using the following expression:

$$\beta = \begin{cases} \left(\dfrac{I}{C} + H^T H\right) H^T X, N \le n \\[2mm] H^T \left(\dfrac{I}{C} + H^T H\right) H^T X, N > n \end{cases} \tag{7}$$

Where $C$ is the network regularization parameter, which is introduced to improve the generalization performance of the ELM-AE method; $X$ is the input sample matrix; n is

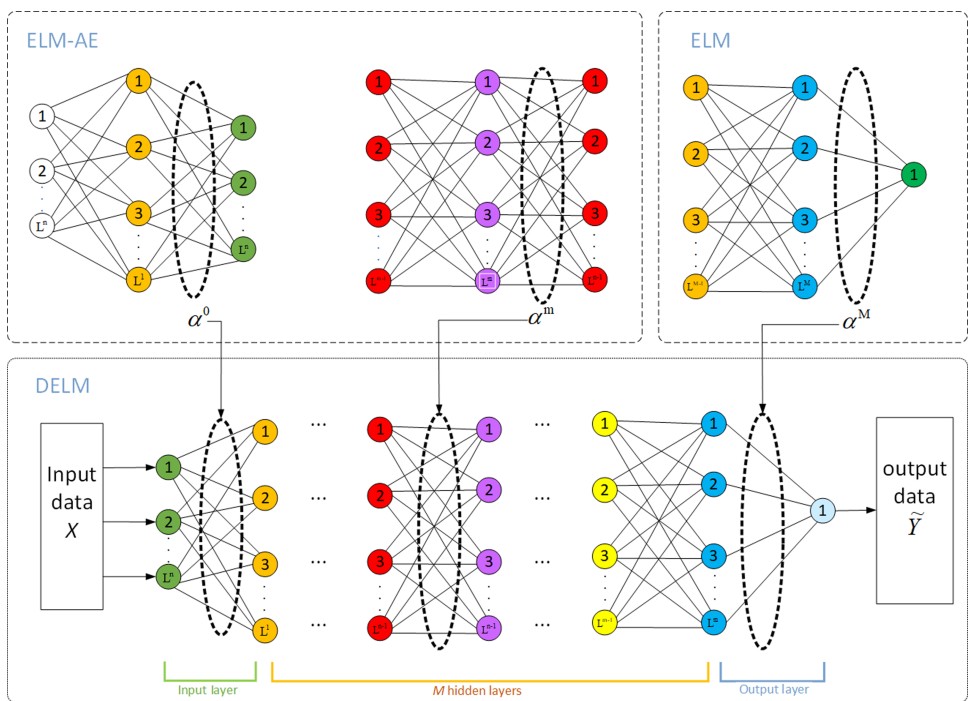

**Figure 6 Model structure of DELM.**

the number of neurons in the hidden layer; N is the number of input samples, and $g()$ is the activation function. By training ELM-AE, unsupervised mapping of samples to depth features is realized.

The input weights of the nodes in each hidden layer are transpositions of the output weights between that layer and the previous layer forming the ELM-AE. Therefore, each layer can be implemented to extract the features of the previous layer. This can be expressed as follows:

$$H^k = g(\beta^k)^T H^{k-1}, k > 1. \tag{8}$$

Unlike other deep learning methods, DELM does not require fine-tuning. Both ELM-AE and the final DELM regression layers use the least-squares multiplication method and only one step of inverse calculation to obtain the updated weights. Consequently, DELM has fast training so it is an appropriate model for online modeling and real-time prediction of temperature fields.

In the present study, the DELM network is constructed using three hidden layers. It should be indicated that the higher the number of hidden layers, the more complex the network, and the higher the training time. To accurately predict temperature distribution, trial simulations were carried out and the number of nodes in the three hidden layers is set to 30, 50, and 50, respectively. Moreover, a tanh function was used as the network activation function.

The following 11 variables were selected as inputs: $x$-coordinate, $y$-coordinate, $z$-coordinate, load increment, total air volume increment, total coal volume increment,

total primary air volume increment, total secondary air volume increment, secondary air temperature increment, coal mill increment (relative position of the flame center), and angle increment. In return, the difference in 3D temperature distribution between the predicted working conditions and the benchmark working condition was the output variable. After IDEM modeling, the incremental prediction of the temperature distribution for different working conditions was obtained separately. The actual temperature field can be obtained by superimposing the incremental output and the benchmark working condition.

## RESULTS & DISCUSSION

### Evaluation indicator

In this article, the IDELM algorithm was used to model six subclasses and obtain the temperature distribution library of this unit under different operating conditions. The IDELM-based prediction results are compared with the values obtained from CFD simulation and ELM, deep belief network (DBN), and deep neural network (DNN) algorithms. The generalization performance and prediction accuracy of algorithms are evaluated using the mean absolute error (MAE), symmetric mean absolute percentage error (SMAPE), and decision coefficient ($R^2$). These indicators are defined as follows:

$$MAE = \frac{1}{N} \sum_{i=1}^{N} |y_p(i) - y_c(i)| \tag{9}$$

$$SMAPE = \frac{1}{N} \sum_{i=1}^{N} \frac{|y_p(i) - y_c(i)|}{(y_p(i) + y_c(i))/2} \tag{10}$$

$$R^2 = 1 - \frac{\left[\sum_{i=1}^{N} (y_p(i) - y_c(i))^2\right] / N}{\left[\sum_{i=1}^{N} (\overline{y}_p(i) - y_c(i))^2\right] / N}. \tag{11}$$

Where $N$ is the number of samples in the test set; $\overline{y}_p$ represents average value of temperature 3D distribution, $y_c$ is numerical simulation value of temperature 3D distribution; $y_p$ indicates prediction of temperature 3D distribution value.

The MAE indicator reflects the average deviation degree between the CFD simulation data and the predicted data. Moreover, the SMAPE indicator is used to evaluate the goodness-of-fit of the model. The lower the values of MAE and SMAPE, the better the prediction performance. Finally, the $R^2$ represents the matching degree between the predicted and numerical simulation data. The closer its value to 1, the stronger the fitting ability of the model.

### Analysis of different sample sizes

In this article, the random sampling method is used to select representative samples among 2.8 million data in each working condition. The distribution of samples in the three-dimensional furnace is shown in Fig. 7. It is observed that samples cover the whole spatial area of the furnace so the integrity of the information is guaranteed. In the main

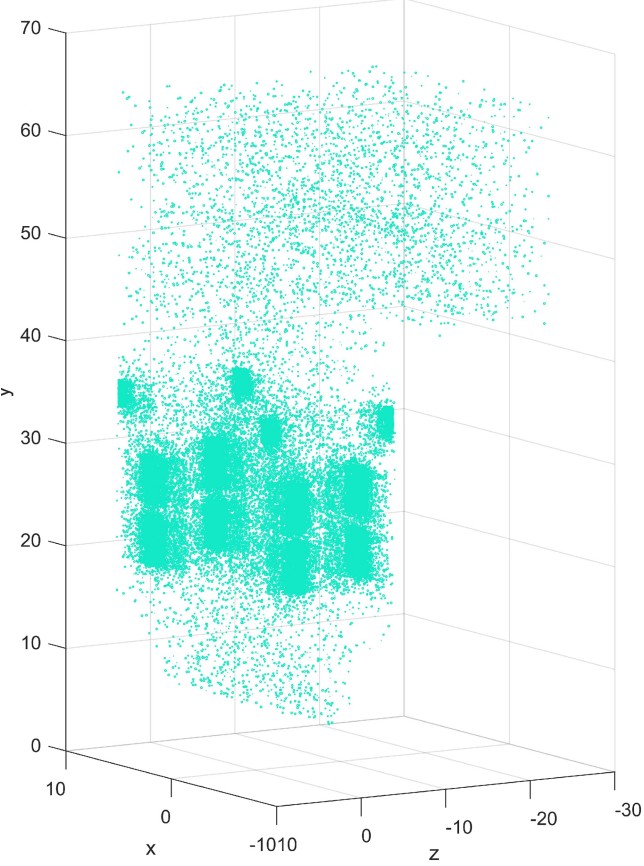

**Figure 7 Scatter plot of samples distribution.**

**Table 7 Results of modeling with different number of samples.**

| Samples | MR2 | MMAE | MSMAPE |
|---|---|---|---|
| 30,000 | 0.84 | 106.50 | 11.75 |
| 50,000 | 0.85 | 103.53 | 11.13 |
| 80,000 | 0.84 | 107.13 | 11.57 |

combustion zone and SOFA zone, the combustion reaction is complex and the temperature distribution varies greatly. Therefore, the selected samples in these zones are relatively dense. Three models with 30,000, 50,000, and 80,000 samples were analyzed respectively. 19 increment-based datasets were used as test sets in turn, the remaining 18 working conditions are randomly sampled, and the datasets were reconstructed. After IDELM modeling, the average values of the performance indexes were calculated. Table 7 shows that the performance indicators of the 50,000 samples are better than those with 30,000 and 80,000 samples. Accordingly, 50,000 samples were selected in all working conditions.

**Table 8** Performance indicators of different algorithms in subclass 2.

| | | DELM | DBN | DNN | ELM |
|---|---|---|---|---|---|
| $R^2$ | max | 0.99 | 0.97 | 0.98 | 0.98 |
| | min | 0.70 | 0.59 | 0.69 | 0.68 |
| | mean | 0.86 | 0.83 | 0.85 | 0.85 |
| MAE | max | 141.53 | 146.28 | 143.49 | 144.62 |
| | min | 26.85 | 32.47 | 31.84 | 34.59 |
| | mean | 96.85 | 106.39 | 103.70 | 99.15 |
| SMAPE | max | 13.58 | 15.62 | 14.08 | 15.38 |
| | min | 2.87 | 4.51 | 3.73 | 3.92 |
| | mean | 10.22 | 13.56 | 10.65 | 11.05 |

## Comparative analysis of different algorithms

To verify the effectiveness of the temperature distribution model based on the proposed method, the DELM prediction results were compared with the results obtained from deep belief network (DBN), deep neural network (DNN), and ELM algorithms. The prediction results in subclass 2 are presented in Table 8. It is observed that the prediction results of the DELM-based model outperform the other models. The mean $R^2$ of the DELM-based prediction model is 0.86, indicating that the prediction model can accurately predict the temperature distribution in the furnace. On the other hand, the DELM-based prediction model has the smallest MAE value, and its SMAPE value is around 10%. Accordingly, it is concluded that the DELM-based temperature distribution prediction model has promising prediction accuracy and excellent generalization ability.

Figure 8 shows the error boxplot of the studied algorithms. It is observed that among the studied algorithms, the DELM algorithm has the lowest absolute error while having a tighter variation bandwidth. The variation of the predicted results using the DELM algorithm is consistent with that of the CFD simulation. It is concluded that the DELM-based model has a reasonable fitting effect and prediction ability.

The PB4 working condition is closest to the average performance of subclass 2. Figure 9 illustrates the PB4 prediction results of four algorithms. We can see that the prediction results of the four algorithm models are consistent with the trend of the CFD target value. The predicted value under the IDELM model is closer to CFD target value, and the results of the other three prediction models of temperature distribution are higher than the target data, and the DBN results have the most deviation.

## Prediction analysis of different working conditions

To analyze the overall performance of the proposed model, the prediction results of the six subclasses are validated respectively. Table 9 shows that the mean $R^2$ of all subclasses is higher than 0.82, which has a good consistency with experimental data. Furthermore, it is found that Subclass 2 has the best model prediction with the highest average $R^2$ and the smallest MAE and SMAPE values. On the other hand, Subclass 3 has the smallest mean $R^2$ and Subclass 5 has the largest prediction error with a mean MAE of 109.76 and SMAPE of 11.02%. It is inferred that as the working condition is closer to the center of clustering,

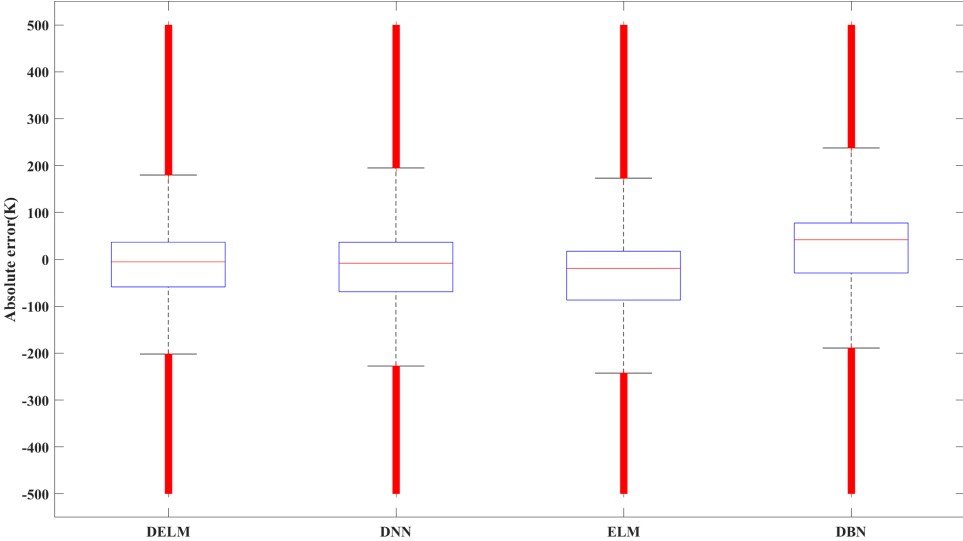

**Figure 8** **Error box plot of different algorithms.**

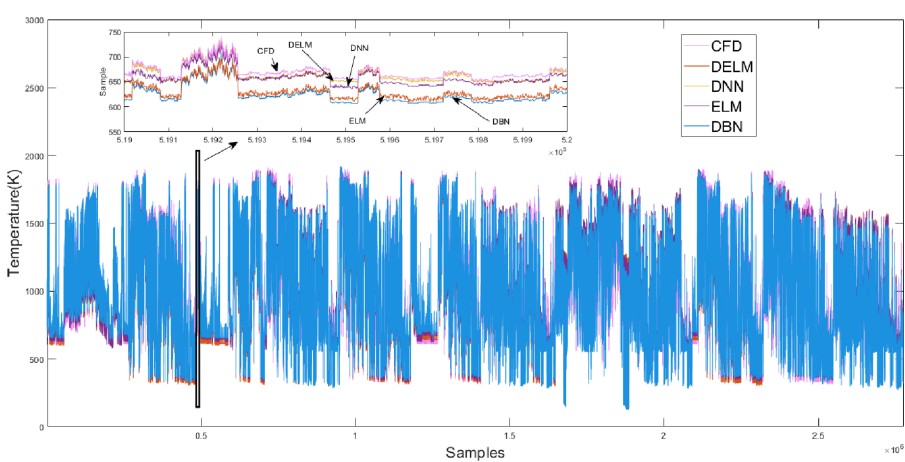

**Figure 9** **Prediction results of different algorithms on the PB4 dataset.**

the prediction results improve and all poor predictions occur near 50% load. The furnace flame filling degree at a lower load is reduced and the temperature 3D distribution changes significantly relative to the benchmark operating conditions.

## CONCLUSIONS

In the present study, the IDELM model was established to predict the temperature distribution in the furnace using CFD simulation. Based on the obtained results and performed analyses, the main conclusions can be summarized as follows:

(1) Combining CFD simulation data of typical working conditions with data-driven machine learning modeling, the temperature distribution is modeled accurately. This model

**Table 9   Performance indicators of different subclasses.**

|       |      | Subclass1 | Subclass2 | Subclass3 | Subclass4 | Subclass5 | Subclass6 |
|-------|------|-----------|-----------|-----------|-----------|-----------|-----------|
|       | max  | 0.98      | 0.99      | 0.99      | 0.99      | 0.97      | 0.99      |
| $R^2$ | min  | 0.7       | 0.70      | 0.69      | 0.67      | 0.71      | 0.71      |
|       | mean | 0.85      | 0.86      | 0.82      | 0.84      | 0.83      | 0.84      |
|       | max  | 137.17    | 141.53    | 144.14    | 155.36    | 135.66    | 138.68    |
| MAE   | min  | 33.19     | 26.85     | 27.56     | 25.34     | 47.76     | 32.14     |
|       | mean | 98.84     | 96.85     | 102.04    | 103.26    | 109.76    | 103.53    |
|       | max  | 15.78     | 13.58     | 18.06     | 18.93     | 18.35     | 15.9      |
| SMAPE | min  | 3.75      | 2.87      | 3.61      | 3.25      | 8.34      | 3.4       |
|       | mean | 11.14     | 10.22     | 12.80     | 12.61     | 12.97     | 11.02     |

can be used in online reconstruction and visualization of the temperature distribution in the furnace.

(2) Special treatments are used to reconstruct data sets. Discrete variables are used to classify typical working conditions and the K-means clustering method is used to set the benchmark conditions. Meanwhile, random sampling is applied to extract representative samples.

(3) Compared with DNN, DBN and ELM algorithms, the IDELM algorithm combines the advantages of deep learning and extreme learning machine to achieve data-driven modeling of temperature distribution at a relatively fast speed. The model accuracy can reach a mean $R^2$ of 0.84, a mean MAE of 102.38, and a mean SMAPE of 11.79%.

Based on the obtained results and the performed analyses, it is concluded that the proposed model can be used to accurately predict temperature distribution and optimize boiler combustion.

## ACKNOWLEDGEMENTS

We thank the review experts for their review.

### Funding

This work was supported by the Jilin Science and Technology Project under grant 20200401085GX. There was no additional external funding received for this study. The funders had no role in study design, data collection and analysis, decision to publish, or preparation of the manuscript.

### Grant Disclosures

The following grant information was disclosed by the authors:
Jilin Science and Technology Project:  20200401085GX.

### Competing Interests

Tao Shen is an employee of Harbin Boiler Company Limited

## Author Contributions

- Manli Lv conceived and designed the experiments, performed the experiments, analyzed the data, performed the computation work, prepared figures and/or tables, authored or reviewed drafts of the article, and approved the final draft.
- Jianping Zhao conceived and designed the experiments, authored or reviewed drafts of the article, and approved the final draft.
- Shengxian Cao conceived and designed the experiments, authored or reviewed drafts of the article, and approved the final draft.
- Tao Shen performed the experiments, analyzed the data, prepared figures and/or tables, and approved the final draft.
- Zhenhao Tang performed the computation work, prepared figures and/or tables, and approved the final draft.

## Data Availability

The raw modeling data and the original data are available in the Supplementary File and at figshare:

Lv, Manli (2023): raw data of error. figshare. Dataset. https://doi.org/10.6084/m9.figshare.21828354.v2.

Lv, Manli (2023): raw data of prection comparison. figshare. Dataset. https://doi.org/10.6084/m9.figshare.21835053.v1.

Lv, Manli (2023): PB4 workong condiction. figshare. Dataset. https://doi.org/10.6084/m9.figshare.21915687.v1.

## Supplemental Information

Supplemental information for this article can be found online at http://dx.doi.org/10.7717/peerj-cs.1218#supplemental-information.

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
