# Peer review of "Prediction of temperature distribution in a furnace using the incremental deep extreme learning machine"

_PeerJ Computer Science, doi:10.7717/peerj-cs.1218_

## Round 0.1 · original submission · Major Revisions

The paper is well written with significant topic. The modelling process is clearly described however the simulation part needs to be improved as the simulation data has not been explained well. In addition, the benefits of data-driven modelling should be summarised to highlight the contributions. A major revision is necessary and a proof reading is recommended.

Reviewer 1 ·

Basic reporting

1)It is noted that your manuscript needs careful editing and pays particular attention to English grammar, spelling, and sentence structure so that the goals and results of the study are clear to the reader.
2)Some Table information didn’t have are not standard forms.
3)The structure of the article should be adjusted. I personally believe that after the introduction of study object, “CFD simulation” should be researched before the “Framework of IDEM modeling”, because this is the basis for all subsequent research

Experimental design

1)In this study, 120 CFD simulation data sets are divided into 6 subclasses. Is it necessary to select a benchmark working condition for each subclass?
2)In this paper, several modeling methods are compared, but the modeling accuracy is not very high and lacks the precision comparison with other studies.

Validity of the findings

Seen from raw data sets, modeling data sets derived from CFD simulation,Since the temperature field distribution can be obtained by CFD simulation, what is the significance of data-driven modeling?

Additional comments

NO additional comments

Reviewer 2 ·

Basic reporting

1)This manuscript needs to pay attention to the normalization of symbols, especially in the parts of “comparative analysis of different algorithms” and “Conclusions”.
2) Formulas (2) and (3) in this paper describe the coal mill operation mode is converted into a flame center position correction factor M. References are preferred here.
3)In figure 8, boxplot and its horizontal and vertical coordinates are not clear enough.

Experimental design

1) Whether the process variables used for modeling can be obtained online?
2) Incremental deep extreme learning machine (IDELM) algorithm was used for modeling in this paper. Did the comparison algorithms of DBN, DNN and ELM also adopt the incremental form?

Validity of the findings

no comment

Additional comments

No additional comments

Reviewer 3 ·

Basic reporting

1)It is noted that your manuscript needs careful editing and pays particular attention to English spelling.
For examples: R2 is an evaluation indicator, but it's written as “R2” a lot in this text
2)In line 49 “including CFD simulation and indirect methods”,Whether this expression is rigorous or not
3)Since deep learning are a point of study in the manuscript, an small introduction about the application of various deep learning algorithms in combustion process should be introduced in background information.

Experimental design

1)Incremental deep extreme learning machine (IDELM) algorithm in the title is described,but the algorithm introduction in subsequent articles is DELM algorithm. Why is there any inconsistency?
2)In this study, data-driven modeling of 3D distribution of temperature field is a new attempt, but the data used is derived from CFD simulation results. As far as I know, CFD itself can get a 3D distribution of temperature. What is the biggest significance of data-driven modeling?
3)Selecting clustering center in each subclass, every clustering center is the 10th working condition of the 20 working condition (BB10 PB10, BP10, PP10, BW10, PW10). is this a coincidence or what's the implicit rules?

Validity of the findings

It can be seen from raw data that this study is based on CFD simulation results of 120 working conditions data was carried out in the study. Can these data truly represent the actual power plant operating conditions?

Additional comments

I have not additional comments about this article.

---

## Round 0.2 · accepted · Accept

All the concerns have been addressed well in the revised version while both reviewer satisfied the revision. Thus the manuscript is ready for publication.

Reviewer 1 ·

Basic reporting

no comment

Experimental design

no comment

Validity of the findings

no comment

Reviewer 3 ·

Basic reporting

The revised version has been revised according to the review comments and is recommended for publication.

Experimental design

The revised version has been revised according to the review comments and is recommended for publication.

Validity of the findings

The revised version has been revised according to the review comments and is recommended for publication

Additional comments

No other comments.